# Adverse birth outcome and associated factors among diabetic pregnant women in Ethiopia: Systematic review and meta-analysis

Demeke Mesfin Belay[1]*, Wubet Alebachew Bayih[1], Abebaw Yeshambel Alemu[1], Aklilu Endalamaw Sinshaw[2], Demewoz Kefale Mekonen[1], Amare Simegn Ayele[3], Wasihun Hailemichael Belayneh[4], Henoke Andualem Tegared[4], Biniam Minuye Birihane[1]

1 Department of Nursing, College of Health Sciences, Debre Tabor University, Debre Tabor, Ethiopia, 2 Department of Pediatrics and Child Health Nursing, School of Health Sciences, College of Medicine and Health Sciences, Bahir Dar University, Bahir Dar, Ethiopia, 3 Department of Midwifery, Reproductive health, Debre Tabor University, Debre Tabor, Ethiopia, 4 Department of Medical Laboratory, Molecular Biology and Immunology, College of Health Sciences, Debre Tabor University, Debre Tabor, Ethiopia

* demekemesfin65@yahoo.com

**Data Availability Statement:** All relevant data are within the manuscript.

## Abstract

### Background

The magnitude of adverse birth outcome among diabetic pregnant women is high in low-and-middle income countries, like Ethiopia. Precise epidemiological evidence is necessary to plan, evaluate and improve effective preventive measures. This systematic review and meta-analysis is the first to estimate the pooled prevalence of adverse birth outcome and associated factors among diabetic pregnant women in Ethiopia.

### Methods

PubMed, Cochrane Library, Google Scholar, SCOPUS, Web of Science and PsycINFO, and article found in University online repository were accessed. Observational studies such as cross-sectional, case-control and prospective cohort reported using English language was involved. $I^2$ statistic was used to check heterogeneity. Egger's test and funnel plot were used to measure publication bias. Weighted inverse variance random effects model was also performed.

### Results

Seven studies with 1,225 study participants were retrieved to estimate the pooled prevalence of adverse birth outcome and associated factors. The pooled prevalence of adverse birth outcome among diabetic pregnant women was 5.3% [95% CI; 1.61, 17.41]. Fasting blood glucose level above 100 mg/dl [Adjusted Odds ratio (AOR) = 10.51; 95% Confidence Interval (CI) = 5.90, 15.12], two hour post prandial glucose level above 120 mg/dl [AOR = 8.77; 95% CI = 4.51, 13.03], gestational age <37 completed week [AOR = 9.76; 95% CI = 5.29, 14.23], no ANC follow-up [AOR = 10.78; 95% CI = 6.12, 15.44], history of previous adverse outcomes [AOR = 3.47; 95% CI = 1.04, 5.90], maternal age < 30 years [AOR =

**Funding:** The author(s) received no specific funding for this work.

**Competing interests:** The authors have declared that no competing interests exist.

**Abbreviations:** ADA, American Diabetes Association; ANC, Antenatal Care; AOR, Adjusted Odd Ratio; CI, Confidence Interval; DM, Diabetes Mellitus; IADPSG, International Association of Diabetes and Pregnancy Study Group; JBI, Joanna Briggs Institute; LMICs, Low and Middle Income Countries; PRISMA, Preferred Reporting Items for systematic review and meta-analysis; WHO, World Health Organization.

3.47; 95% CI = 1.04, 5.90], and illiteracy [AOR = 2.89; 95% CI = 0.81,4.97)] were associated factors of adverse birth outcome.

## Conclusions

The pooled prevalence of adverse birth outcomes among diabetic pregnant women in Ethiopia was high. Child born from mothers who were illiterate, maternal age < 30 years, gestational age < 37 completed weeks, history of previous adverse birth outcomes and no ANC follow-up increased the risk of adverse birth outcome.

## Trial registration

It is registered in PROSPERO data base: (PROSPERO 2020: CRD42020167734).

## Introduction

Diabetes Mellitus (DM) is a group of metabolic disorders characterized by a high blood sugar level over a prolonged period of time and caused by either from deficiency in insulin secretion, decreased insulin action or both [1]. Along with other form of DM; child-bearing women are at a higher risk of developing DM in pregnancy [1, 2]. This result in hyperglycemia in pregnancy is a medical condition resulting from either pre-existing diabetes or gestational diabetes which increased the risks of adverse birth outcomes [3]. DM in pregnancy was high (90%) in Low and Middle Income Countries (LMICs) where access to maternal and child health service is limited.

DM in pregnancy can be gestational DM; affects 2–3% of pregnancy [4] and pre-existing DM; affects 0.2-.0.3% pregnancy [5]. Accordingly, DM in pregnancy affects 17% of pregnancy [6]. Therefore, a woman with hyperglycemic pregnancy have higher chance of developing adverse birth outcomes, like congenital anomaly, prematurity, still birth, macrosomia, neonatal hypoglycemia and spontaneous abortion, regardless of major improvement in clinical management [7–9]. Although the magnitude and burdens of DM in pregnancy is high in LMICs, including Ethiopia, little is known about adverse birth outcomes in these countries [6]. Population based study in twelve regions of Denmark showed that the risk of adverse fetal outcome was higher in a woman with hyperglycemic pregnancy. as compared to the general population [10]. Globally, 75% of neonatal mortality and morbidity is due to adverse birth outcomes [11]. Despite neonatal mortality and morbidity is declined globally; highest in sub-Saharan Africa estimated at 27.7% deaths per 1000 live birth in 2018 [12].

Despite preterm birth has been occurred in general population; the risk is higher among diabetic pregnant women. According to 2016 World Health Organization (WHO) report, the risk of preterm birth from diabetics mother were increased by 5% in LMICs [13]. Similarly, DM in pregnancy increases the likelihood of macrosomia by 50%, a 3-fold increase as compared with non-diabetic pregnant women [14]. Systematic review by WHO and International Association of Diabetes and Pregnancy Study Groups (IADPSG) identified that DM in pregnancy have 81% higher risk of macrosomia [15]. In addition, systemic review done in sub-Saharan Africa found that the rate of macrosomia from diabetic pregnant women accounts 80% [16]. Currently, the rate of spontaneous abortion accounts 9–14% among diabetic pregnant women. However, the rate of spontaneous abortion rise to 44% when there is poor control of the blood glycemic level and the disease becomes advanced. Besides, the magnitude of congenital anomaly among the general populations accounts 1–2%. But in women with DM,

the risk of congenital anomaly increased by 4–8 folds. Furthermore, around 15–25% of neonates delivered from diabetic pregnant women develop hypoglycemia [17]. Systematic review in LMICs indicated that the incidence of neonatal hypoglycemia among gestational DM was 5.1%-30.4% higher [18]. Similarly, the magnitude of still birth is higher among diabetic mother. Systematic review in LMICs indicated that the incidence of stillbirth was 6.3% higher among diabetics mothers [18]. Lastly, evidence in different part of Ethiopia showed that the magnitude of adverse birth outcomes ranges from 1.42%-8% [19–24].

Evidence advocated that inadequate antenatal, medical, and preconception care are factors affecting adverse birth outcomes among diabetic women [6, 25, 26]. American Diabetes Association (ADA) [27] and International Diabetic Federation (IDF) [28] sets standards of medical cares like preconception counseling and preconception care to prevent the resulting adverse birth outcomes.

Even though, the government of Ethiopia set different plan and strategies to combat this problem; it is still the major public health issue [29]. Studies done to identify magnitude and associated factors of adverse fetal outcomes among diabetic pregnant women in Ethiopia are limited and inconsistent depends on the populations studied and the methodology used. Therefore, this systematic review and meta-analysis is aimed to estimate the pooled prevalence of adverse birth outcomes and associated factors among diabetic pregnant women in Ethiopia.

## Methods

### Reporting

This systematic review and meta-analysis was designated to estimate the pooled prevalence of adverse birth outcomes and associated factors among diabetic pregnant women in Ethiopia. The result is reported based on standard Preferred Reporting Items for Systematic review and Meta-analysis (PRISMA) checklist [30] (S1 Checklist). The review protocol was submitted for registration in the international prospective register of systematic reviews (PROSPERO) and registered with PROSPERO registration number (PROSPERO 2020: CRD42020167734).

### Database and search strategy

International data bases, which are PubMed, Cochrane Library, Google Scholar, SCOPUS, Web of Science and PsycINFO, and article found in Addis Ababa and Haramaya University online repository, were searched. Reference lists and citations of included papers were checked to identify any other potentially relevant papers. Compressive search strategy from March 29/ 2020 to May 26/ 2020 has been employed using Population, Intervention, comparison and Outcomes (PICO) formulating question and search terms ("Neonatal outcomes", "Perinatal outcomes", "Birth outcomes", Fetal outcomes", "Immaturity," "Premature", "Preterm birth", Abnormal birth weight, Still birth, Big baby, Macrosomia, Birth defect, Congenital Anomaly, "Congenital defect," "Hypoglycemia," "Miscarriage," "Abortion," "Pre-gestational diabetes mellitus," "Diabetes mellitus type 1", "Diabetes mellitus type 2", "Gestational Diabetes mellitus", "Hyperglycemia", "Pregnant women", and "Ethiopia"). "AND" and "OR" Boolean operators were used to combined search terms.

### Inclusion and exclusion criteria

Five articles published from year 2013–2019 and two unpublished articles in Addis Ababa and Haramaya University online repository reported in English language was selected for analysis. Observational studies such as cross-sectional, case-control and prospective cohort study reporting the prevalence of adverse birth outcomes or and a minimum of one contributing

factors for adverse birth outcomes conducted in Ethiopia were included. Studies which used WHO diabetes mellitus diagnosis approach were included. Ethiopia has been used WHO diagnosis approach; based on this; diabetes in pregnancy should be diagnosed if one or more of the following conditions were met: (1) fasting plasma glucose greater or equal to 7 mmol/l (126mg/dl); 2-hour plasma glucose greater or equal to 11.1mmol/l (200mg/dl) following a 75 g oral glucose load; (3) random plasma glucose greater or equal to 11.1 mmol/l (200mg/dl) in the presence of diabetes symptoms. However, articles without full-text or abstract, with JBI critical appraisal score less than 50% and not reporting the outcomes of the interest were excluded.

## Study selection and quality assessment

All retrieved studies were exported to EndNote version 9 (Thomson Reuters, London) reference manager and duplicated studies were carefully removed. Two investigator (BM and AS) independently screened the titles and abstracts which were followed by a full text review to determine eligibility of each study. Any difference was resolved by third author (W.A). Two authors (D.M. and B.M.) evaluated independently the eligibility of all retrieved studies by using the Joanna Briggs Institute (JBI) quality appraisal checklist adapted for studies reporting prevalence data, cross-sectional, prospective cohort and case-control studies [31]. The following items were used to review cross-sectional studies: (1) inclusion criteria; (2) description of study subject and setting; (3) valid and reliable measurement of exposure; (4) objective and standard criteria used; (5) identifications of confounder; (6) strategies to handle confounder; (7) outcome measurement; and (8) appropriateness of statistical analysis. The following items were also used for appraising cohort studies: (1) similarity of group; (2) similarity of exposure measurement; (3) validity and reliability of measurement; (4) identifications of confounder; (5) strategies to handle confounder; (6) appropriateness of groups/participants at the start of the study; (7) validity and reliability of outcome measured; (8) sufficiency of follow up time; (9) completeness of follow up or descriptions of reason to loss to follow-up; (10) strategies to address incomplete follow-up; (11) appropriateness of statistical analysis. Moreover, the following items were used for appraising case-control studies: (1) comparable group; (2) appropriateness of case and control; (3) criteria to identify case and control; (4) standard measurement of exposure; (5) similarity in measurement of exposure for case and control; (6) handling of confounder; (7) strategies to handle confounder; (8) standard assessment of outcome; (9) appropriateness of duration for exposure; and (10) appropriateness of statistical analysis. Studies considered low risk whenever fitted to 50% and or above quality assessment check list criteria's.

## Data extraction

After collecting the required finding form the entire databases, all important data were extracted by authors (A.S, and A.Y.) using standardized JBI data extraction form and cross cheeked to ensure consistency. Three independent authors (D.M, W.A, and B.M.) extract data on author/s name, year of publication/study, study area/region, study design, sample size, prevalence of adverse fetal outcome with 95% CI and associated factors were collected. Any dissimilarity and inconsistencies were resolved among the authors by discussion and repeating the procedure. The reviewer contacted the corresponding author(s) for further information whenever pertinent data were missed from the included studies.

## Outcomes of measurement

Adverse birth outcomes: The presence of at least one of the adverse outcome (stillbirths, congenital anomaly, neonatal hypoglycemia, spontaneous abortion, macrosomia and preterm birth).

## Data analysis

The extracted data were transferred to Stata version 14 statistical software for meta-analysis. Meta-analysis of prevalence of adverse birth outcome was determined using random effects model [32] to result a pooled effect size with 95% CI. The estimated effect of selected independent factors was analyzed and presented using forest plot. Measure of association using AOR with 95% CI was reported. Heterogeneity of the study was evaluated using Cochrane Q-test and I-squared statistics. I-squared was used to calculate the percentage of total variation in the study estimated due to heterogeneity. I-squared range between 0 and 100%; the values of $I^2$ 25, 50, and 75% represented low, moderate and high heterogeneity respectively. A p-value of less than 0.05 was used to declare significant heterogeneity [32, 33]. The random effects model using Der Simonian and Laird method is the most common method in a meta-analysis to adjust for the observed variability.

Funnel plot and Egger's regression test was used to check the presence of publication bias [34]. We also employed Egger's and the Begg's test to determine if there was significant publication bias. A p-value of less than 0.05 was considered significant publication bias [35]. Finally, we performed a sensitivity analysis to describe whether the pooled effect size was influenced by individual studies.

## Results

### Search result and study characteristics

Three hundred seventy seven articles were retrieved using a search strategy from PubMed, Cochrane Library, Google Scholar, SCOPUS, Web of Science, PsycINFO, Addis Ababa and Haramaya University online repository. Three hundred twenty five articles were remained after 50 duplicated articles were obliterated. From the remaining 325 articles, 255 articles were excluded after review of the titles and abstracts. Out of the remaining 70 articles, 32 articles were omitted because their full text was not available. Finally, 38 articles undertook for full text selection. 31 articles were excluded based on the predetermined eligibility criteria. Finally, 7 articles were included for analysis (Fig 1).

A total of 7 studies with 1,225 study participants were included in this systematic review and meta- analysis. From 7 studies four were done in Addis Ababa [19, 22–24], one in Amhara region [36], and the remaining two in Oromoia region [20, 37]. All studies were conducted at different hospitals of Ethiopia using different study design; four cross-sectional [20, 22–24], two case-controls [36, 38], and one prospective cohort [19]. This systematic review and meta-analysis also showed factors associated with adverse birth outcome among diabetes women. Those are socio-demographic factors (being house wife, maternal age less than 30 years old, rural residency, and illiteracy), obstetrics factor (preconception care, Gestational age less than 37 completed weeks, previous history of adverse birth outcome, no antenatal care follow up, and short birth spacing) and medical related factors (presence of glucometer at home, average fasting blood glucose level, and average 2 hour post prandial glucose level) (Table 1).

### Quality of the included studies

One study was assessed using JBI checklist for prospective Cohort [19], four studies using JBI checklist for cross-sectional studies [20, 22–24] and two studies using JBI checklist for case-control studies [36, 38]. None of the studies were excluded based on the quality assessment criteria's.

### Pooled meta-analysis of adverse birth outcome

Seven studies with 1,225 participants were retrieved. The overall pooled prevalence of adverse birth outcome among diabetic women in Ethiopia was 5.3% [95% CI; 1.61, 17.41; $I^2$ = 0.0%, P = 0.656] (Fig 2). No heterogeneity ($I^2$ = 0.0%, p = 0.656) observed.

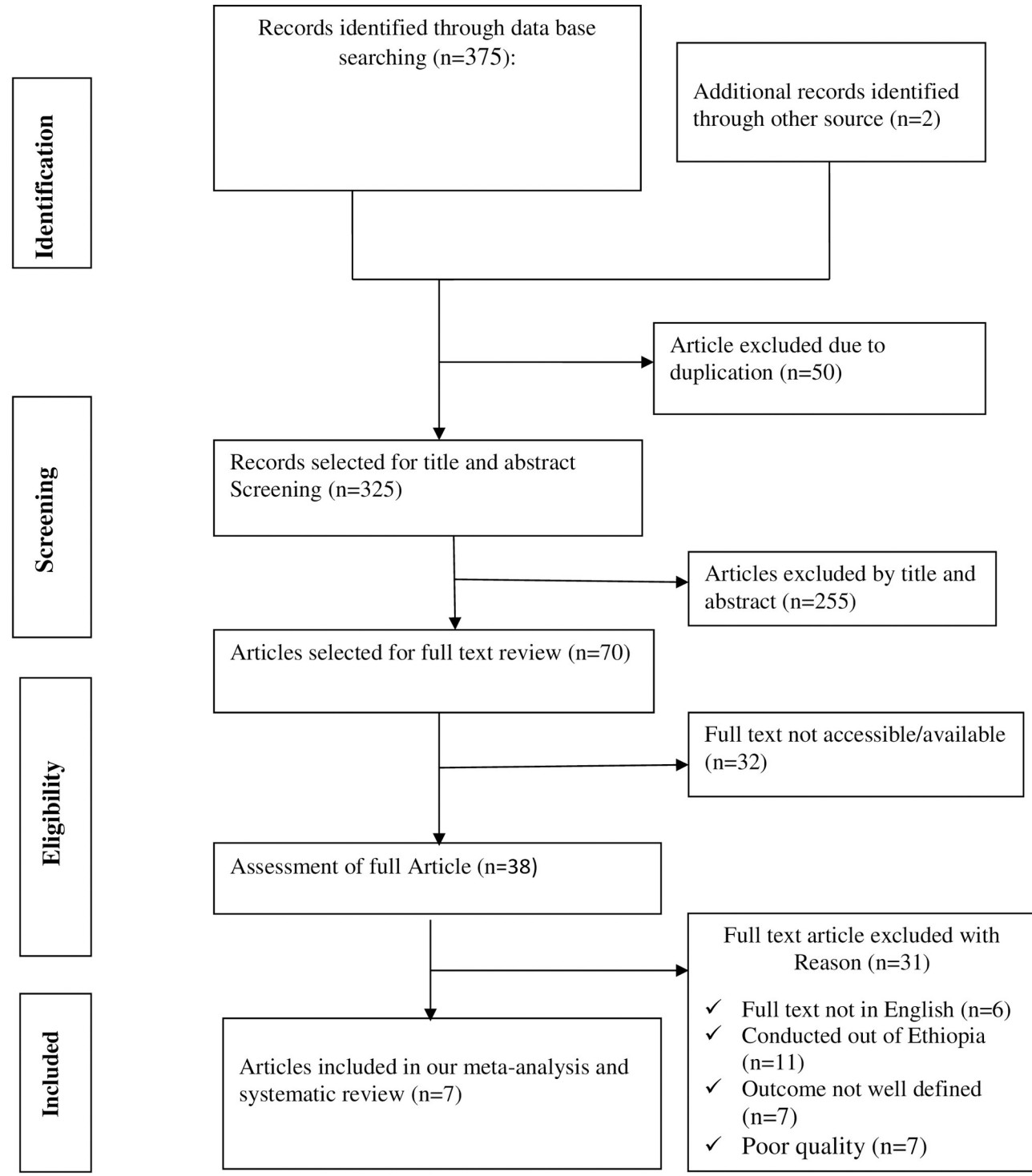

**Fig 1. Flow diagram of study selection for systematic review and meta- analysis of adverse birth outcomes and its associated factors among diabetic women in Ethiopia, 2020.**

**Table 1. Study characteristics included in systematic review and meta-analysis of adverse birth outcome and its associated factors among diabetic pregnant women in Ethiopia, 2020 1000262261674.**

| Authors | Regions | Area | Study design | Sample size | Prevalence | Response rate | Outcome variable | Associated factors | Quality |
|---|---|---|---|---|---|---|---|---|---|
| Talema A et al. [19]. | Addis Ababa | Teaching hospital in Addis Ababa | Prospective cohort | 80 | - | 100% | Adverse birth outcome | Glucometer at home & Preconception care | Low risk |
| Elias B et al. [38]. | Oromoia | Hiwot Fana & Dilchora Hospital | Un matched case-control | 45 | 31% | 100% | Adverse birth outcome | - | Low risk |
| Bajrond E et al. [23]. | Addis Ababa | Tikur Anbessa Hospital | Retrospective cross-sectional | 337 | 18% | 100% | Adverse birth outcome | Being house wife, preterm delivery | Low risk |
| Selamawit E et al. [24]. | Addis Ababa | Tikur Anbessa Hospital | Retrospective cross-sectional | 162 | 78.40% | 80.20% | Adverse birth outcome | Average FBG, average 2 hr pp & Low maternal age | Low risk |
| Abdisa B et al. [20]. | Oromoia | Mettu Karl Hospital | Retrospective cross-sectional | 346 | 17.60% | 95.60% | Adverse birth outcome | Being house wife, preterm deliver | Low risk |
| Abay W et al. [36]. | Amhara | Desie, Debre Birhan, & Bahir Dare Referal hospital | Un matched case-control | 134 | - | 97.10% | - | Rural, illiteracy, no ANC, previous adverse birth outcome & short birth spacing | Low risk |
| Zewedu G [22]. | Addis Ababa | Selected hospital in Ethiopia | Cross-sectional | 111 | 4.50% | 100% | Adverse birth outcome | Low maternal age | Low risk |

"FBG": Fasting Blood Glucose; "PP": Post prandial.

## Publication bias

Visual inspection of the funnel plot was used to assess asymmetry distributions of the adverse birth outcome (Fig 3). Egger's regression test showed that the presence of publication bias at p = 0.013 (Table 2).

## Trim and fill analysis of pooled prevalence of adverse birth outcomes

Trim and fill analysis was done, three studies were added and the total number of the studies become 10. The pooled prevalence of adverse birth outcomes is 1.2 at p-value 0.052 (Table 3).

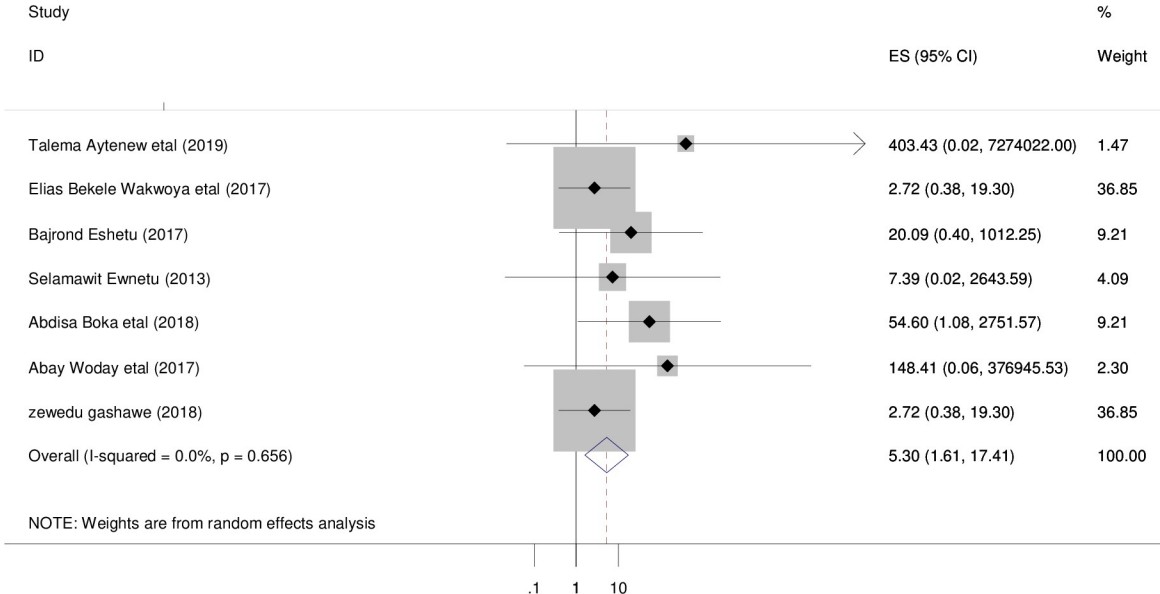

**Fig 2. Forest plot of overall prevalence of adverse birth outcomes among diabetic women in Ethiopia, 2020.**

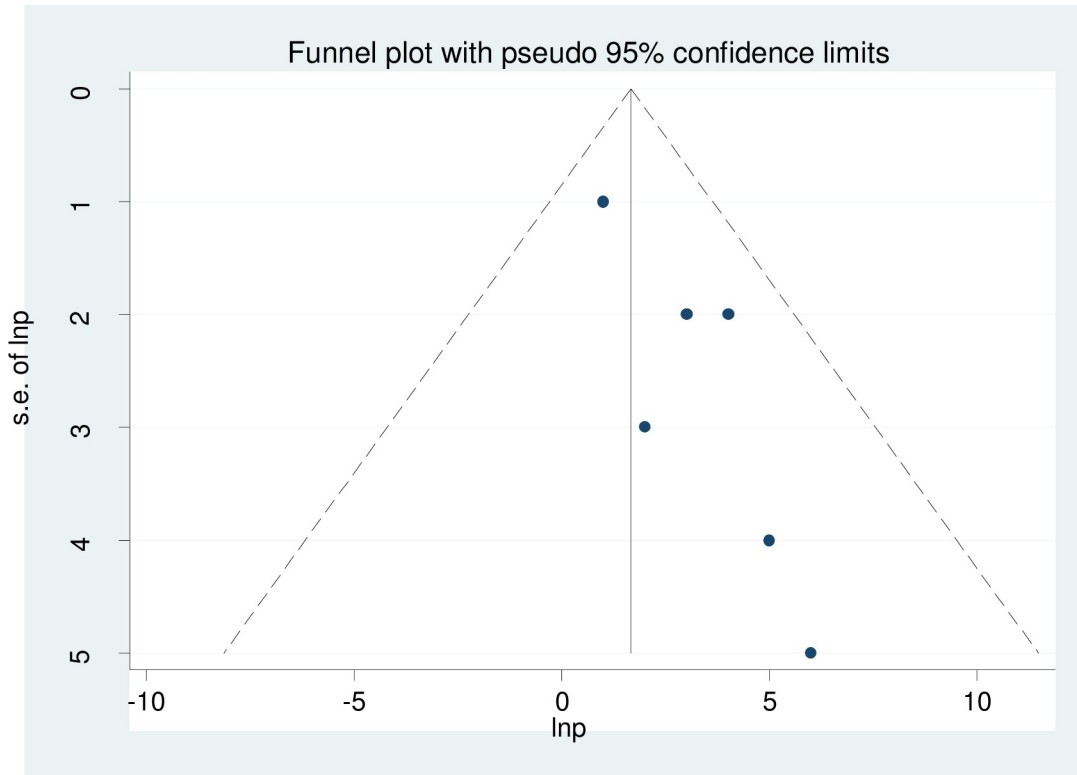

**Fig 3. Funnel plot to show publication bias.**

### Investigations of heterogeneity

Given that the result of this meta-analysis revealed no a statistically significant heterogeneity among studies to show the overall pooled prevalence of adverse birth outcome ($I^2 = 0.0\%$, p = 0.656).

### Sensitivity analysis

The result of sensitivity analyses using random effects model suggested that Talema A et al. influenced the overall estimate significantly (Fig 4).

### Factors associated with adverse birth outcome

In this systematic review and meta-analysis, socio-demographic factors, obstetrics factors and maternal medical illness related factors contributing for adverse birth outcome among diabetic women (Table 4).

### Medical illness related factors

Study done in Tikur Anbessa Specialized Hospital, Addis Ababa showed that fasting blood glucose level above 100mg/dl [AOR = 10.51 (95% CI; 5.90, 15.12)], and average 2 hour post

**Table 2. Egger's test to show publication bias for each adverse birth outcomes among diabetic pregnant women in Ethiopia, 2020.**

| Outcomes | Std_Eff | Coef. | Std. Err. | T | P>|t| | 95% CI |
|---|---|---|---|---|---|---|
| Overall adverse birth outcomes | Slope | -.20 | .56 | -0.35 | 0.74 | -1.65 1.25 |
| | Bias | 1.34 | .35 | 3.81 | 0.01 | 4351183 2.24 |

**Table 3. Trim and fill analysis of overall pooled prevalence of adverse birth outcomes among diabetic pregnant women in Ethiopia, 2020.**

| Meta-analysis | | | | | | |
|---|---|---|---|---|---|---|
| Method | Pooled est. | 95% CI | | Asymptotic | | No. of studies |
| | | Lower Upper | | z-value p-value | | |
| Fixed | 1.39 | 0.14 | 2.63 | 2.19 | 0.03 | 7 |
| Random | 1.39 | 0.14 | 2.64 | 2.19 | 0.03 | |

Test for heterogeneity: Q = 2.223 on 6 degrees of freedom (p = 0.898)

Moment-based estimate of between studies variance = 0.000

Trimming estimator: Linear

Meta-analysis type: Fixed-effects model

| Iteration | Estimate | Tn | # to trim | Diff | | |
|---|---|---|---|---|---|---|
| 1 | 1.385 | 25 | 3 | 28 | | |
| 2 | 1.200 | 25 | 3 | 0 | | |

Filled

| Meta-analysis | | | | | | |
|---|---|---|---|---|---|---|
| Method | Pooled est. | 95% CI | | Asymptotic | | No. of studies |
| | | Lower Upper | | Z-value P-value | | |
| Fixed | 1.20 | -0.01 | 2.41 | 1.94 | 0.05 | 10 |
| Random | 1.20 | -0.01 | 2.41 | 1.94 | 0.05 | - |

Test for heterogeneity: Q = 4.015 on 9 degrees of freedom (p = 0.910)

Moment-based estimate of between studies variance = 0.000

prandial glucose level above 120mg/dl [AOR = 8.77 (95% CI; 4.51, 13.03)] increase the odds of adverse birth outcome among diabetic pregnant women as compared with that of non-diabetic women.

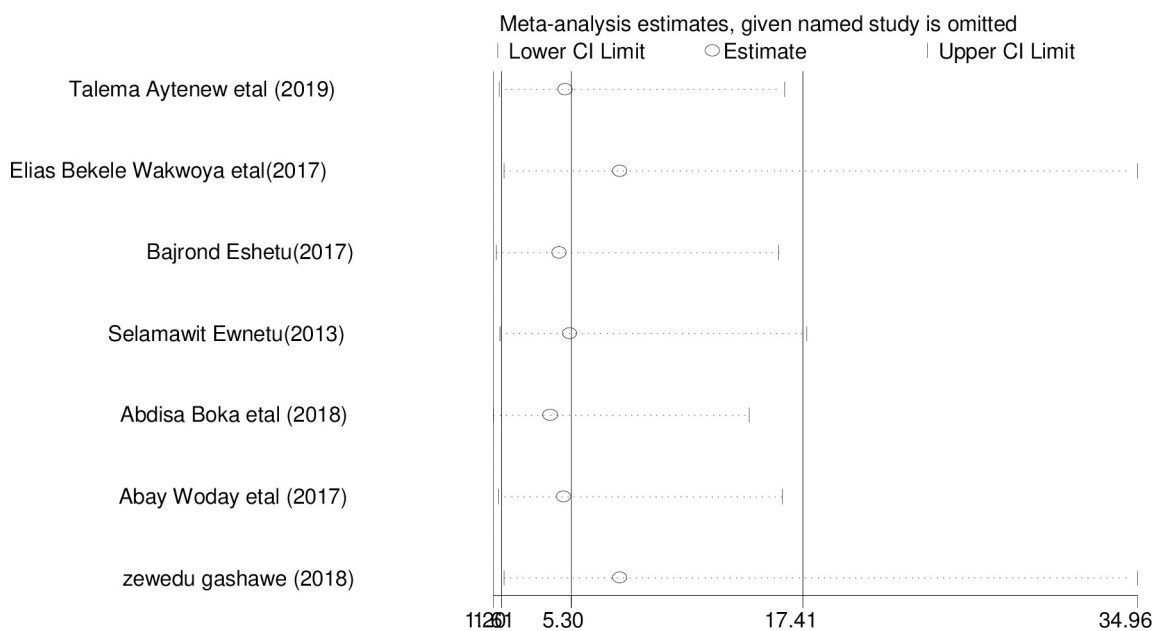

**Fig 4. Sensitivity analysis of adverse birth outcomes among diabetics mother in Ethiopia, 2020.**

**Table 4. Summary of associated factors with adverse birth outcomes among diabetic pregnant women in Ethiopia, 2020.**

| Variables | Model | Publication bias & Egger's test | Status of heterogeneity | AOR | I² | P-value |
|---|---|---|---|---|---|---|
| GA <37 week | Random | 0.19 | Significant | 9.76 | 86.30% | 0.00 |
| No ANC follow up | Random | 0.19 | Significant | 10.78 | 86.30% | 0.00 |
| Previous adverse birth outcome | Random | 0.19 | Significant | 3.47 | 86.30% | 0.00 |
| Average FBG good | Random | 0.04 | Significant | 10.51 | 100% | - |
| Average 2 hour PP blood glucose | Random | 0.04 | Significant | 8.77 | 100% | - |
| Maternal age <30 years | Random | 0.00 | Moderate | 3.47 | 69.30% | 0.01 |
| Unable to read and write | Random | 0.00 | Moderate | 2.89 | 69.30% | 0.01 |

"GA": Gestational Age; "FBG": Fasting Blood Glucose; "PP": Post prandial.

### Obstetrics related factors

Different studies done in different hospital of Addis Ababa indicated that the odds of having adverse birth outcome was 9.76, 10.78, and 3.47 times higher among diabetic pregnant women with gestational age less than 37 completed week [AOR = 9.76 (95% CI; 5.29, 14.23)], no ANC follow up [AOR = 10.78 (95% CI; 6.12, 15.44)], and previous adverse outcome [AOR = 3.47 (95% CI; 1.04, 5.90)] respectively.

### Socio-demographic factors

Two different studies in Tikur Anbessa Specialized Hospital and one study in selected governmental hospital, Addis Ababa indicated that the odds of having adverse birth outcomes among diabetics women age less than 30 years old was 3.47 [AOR = 3.47 (95% CI; 1.04, 5.90)] times higher as compared with its counterpart. Similarly, a study in Hiwot Fana Specialized University Hospital and Dilchora Hospital showed that the odds of having adverse birth outcome among diabetic illiterate women was 2.89 [AOR = 2.89 (95% CI;0.81,4.97] times higher as compared with its counterpart.

## Discussion

Globally, the prevalence of undiagnosed DM in pregnancy has been rises which in turn increases the magnitude of adverse birth outcome both in pregnant women and live births. The pooled prevalence of adverse birth outcome among diabetic women was 5.3% [95% CI; 1.61, 17.41) higher as compared with non-diabetic mother. This review also showed that factors like fasting blood glucose level above 100mg/dl, average 2 hour post prandial blood glucose level above 120mg/dl, gestational age less than 37 weeks, no ANC follow up, and previous adverse birth outcome increase the odds of adverse birth outcome among diabetic women.

The pooled prevalence of adverse birth outcome among diabetic women in Ethiopia is 5.3% higher as compared with its counterpart [95% CI; 1.61, 17.41]. This is in line with the study done in Kathmandu Medical College Teaching Hospital 8.89% [39], Kulashekaram 13.4% [40], Dharan 10.66% [41], Saudi Arabia 2.7% [37], Denmark 2.5% [10], and China 15 .6% [42], This could be due to the health care package and system towards maternal and newborn health is similar between the countries. In addition, different countries are realizing different strategies and preventive measures to reduce the resulting adverse birth outcome.

But, the finding in Ethiopia is lower than the study done in Trichy SRM Medical College Hospital (70%) [43], Bhumibol Adulyadej Hospital 47.6% [44], Central India 36% [45], and Thailand 20% [46]. This discrepancy could be there is limited capacity to screen hyperglycemia in pregnant women and to identify possible adverse birth outcome in Ethiopia. Beside,

difference in universal screening strategies of DM during pregnancy in Ethiopia (WHO) and the rest has been used (ADA, and IADPSG & etc.). Moreover, it might be due to socio-demographic, environmental and genetically difference of the study populations.

However, the finding form this review is higher than retrospective cohort study conducted in Qatar 1.05% [47]. The possible reason for this difference could be there is early screening of the problems and providing appropriate intervention in Qatar. Furthermore, the study participant in Qatar were those who have ANC follow up and delivered at hospitals in contrast to the study setting. This could be due to socio- economic difference between the two countries.

According to this review the odds of having adverse birth outcome was 10.51 and 8.77 times higher among diabetics women with fasting blood glucose level greater than 100mg/dl and average 2 hour post prandial glucose level greater than 120mg/dl, respectively. This showed that uncontrolled maternal hyperglycemia increase the likely hood of having adverse birth outcome. This is due to the fact that utilizations of glucose by the fetal cells becomes high since maternal hyperglycemia increase fetus blood glucose level which further increase fetal insulin secretions leads to macrosomia. Beside, this is due to poor maternal glycemic control affects different small and large blood vessels result in insufficient circulation or high blood pressure causes poor perfusions of the fetus end up with slow growth of fetus in the uterus result in still birth and prematurity [48]. Moreover, maternal hyperglycemia makes the fetus to secrete more insulin to handle excessive sugar that passes from the mother to the fetus via placenta. After birth, the supply of sugar from the mother through the placenta to the fetus is cut off, but the newborn still secretes extra insulin. This extra insulin brings blood sugar level down too low, causing hypoglycemia in the neonates [49]. Maternal hyperglycemia decrease the level of $HbA_1C$ in the fetus which result in congenital anomaly and spontaneous abortion [50].

Obstetric factors like gestational age less than 37 completed weeks, no ANC follow up, and previous adverse birth outcome significantly associated with adverse birth outcome among diabetic pregnant women as compared with non-diabetic women. The odds of adverse birth outcome was 9.76%, 10.78%, and 3.47% higher among diabetic women with gestational age less than 37 completed weeks, no ANC follow up, and previous adverse birth outcomes respectively. This is due to the fact that ANC follow up is the right tool to provide preconception care like glycemic control, help to reduce maternal complication and fetal complication, supplementations of iron which is important to prevent the resulting congenital anomaly. In general ANC follow up help to control blood glucose level to an optimal level which help to prevent the possible adverse birth outcomes among diabetics women [51]. Complications which occurs during pregnancy affect the well-being of the fetus in the uterus and are more vulnerable for further adverse birth outcomes [23]. Mothers with previous adverse birth outcomes has different risk factors like smoking, alcohol drinking, and undiagnosed chronic medical disease (hypertension and pre-existing DM) which furthers increase the likely hoods of resulting adverse birth outcomes beside to maternal hyperglycemia [52].

The odds of having adverse birth outcome among diabetics women age less than 30 years old [AOR = 3.47 [95% CI; 1.04, 5.90] was 3.47 times higher as compared with its counterpart. It is due to the fact that maternal age 25–34 years old has high body mass index (obese) [53] and lipid metabolism is altered in pregnancy. In third trimester, there is prominent lipolysis promoted by insulin resistance [54] which increase triglyceride which is important fuels for fetal growth [53].

It is better to give great stress for mother with previous history of adverse birth outcome. It is also recommended that providing and strengthening regular and timely focused antenatal care service and early detection and timely maternal glycemic control shall be given a particular emphasis. Moreover, it is better to provide health education for the community about adverse birth outcomes.

## Strength and limitations of the study

This study is the first systematic review and meta-analysis that estimate the national prevalence of adverse birth outcomes and contributing factors among diabetic women. Limited numbers of primary articles are used proportionally in each regions of the country. All the studies are institutional based which affects the representativeness to the general community. In additions, there were no sufficient studies about the possible risk factor for adverse birth outcome.

## Conclusions

According to this systematic review and meta-analysis, the pooled prevalence of adverse birth outcome among diabetic women in Ethiopia was 5.3%. Unable to read and write, maternal age less than 30 years old, gestational age less than 37 completed weeks, previous adverse birth outcomes and no ANC follow up was significantly associated with adverse birth outcomes. So, early screening of diabetes during pregnancy and follow up is important to alleviate the burden of maternal and neonatal adverse birth outcomes. In addition, concern should be given in improving antenatal follow up.

## Supporting information

**S1 Checklist. PRISMA checklist.**
(DOC)

## Acknowledgments

We would like to thank all authors of studies included in this systematic review and meta-analysis.

## Author Contributions

**Conceptualization:** Abebaw Yeshambel Alemu.

**Data curation:** Demeke Mesfin Belay.

**Formal analysis:** Demeke Mesfin Belay, Wasihun Hailemichael Belayneh.

**Investigation:** Wubet Alebachew Bayih, Aklilu Endalamaw Sinshaw.

**Methodology:** Demeke Mesfin Belay, Demewoz Kefale Mekonen, Biniam Minuye Birihane.

**Software:** Demeke Mesfin Belay, Wubet Alebachew Bayih, Demewoz Kefale Mekonen.

**Supervision:** Demeke Mesfin Belay.

**Validation:** Henoke Andualem Tegared.

**Writing – original draft:** Demeke Mesfin Belay, Amare Simegn Ayele.

**Writing – review & editing:** Demeke Mesfin Belay, Amare Simegn Ayele.

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
