## [Decision Letter · Decision Letter 0]

3 Oct 2020

PONE-D-20-26756

Adverse Birth Outcomes and Associated Factors among Diabetic Pregnant Women in Ethiopia; Systematic review and Meta-analysis

PLOS ONE

Dear Dr. Belay,

Thank you for submitting your manuscript to PLOS ONE. After careful consideration, we feel that it has merit but does not fully meet PLOS ONE’s publication criteria as it currently stands. Therefore, we invite you to submit a revised version of the manuscript that addresses the points raised during the review process.

ALL of the reviewer's comments must be addressed in your revised manuscript.

We look forward to receiving your revised manuscript.

Kind regards,

Frank T. Spradley

Academic Editor

PLOS ONE

Reviewers' comments:

Reviewer's Responses to Questions

**Comments to the Author**

1. Is the manuscript technically sound, and do the data support the conclusions?

Reviewer #1: Partly

2. Has the statistical analysis been performed appropriately and rigorously? 

Reviewer #1: Yes

3. Have the authors made all data underlying the findings in their manuscript fully available?

Reviewer #1: Yes

4. Is the manuscript presented in an intelligible fashion and written in standard English?

Reviewer #1: No

5. Review Comments to the Author

Reviewer #1: I would like to congratulate the authors to address such a good topic to discussion. It is so important to bring to light the information about diabetes, which is an underestimated and poor screened disease during pregnancy. It is a condition which is relatively cheap to be detected and when in the majority of the cases have a good control with conservative approach but the implications of not address it is importance could affect negatively mother and children.

All my comments are in order to improve the quality of the paper for the reader.

Tittle

The title is appropriate and brings curiosity for the reader to continue reading.

Abstract

The abstract is appropriate, however, if you agree to change the points of the text, this must also be readjusted in the abstract.

Introduction session

#1 The first paragraph of the introduction is a little bit complicated to understand, I suggest a major review of the English and rewrite it for better read.

#2 In the second paragraph: what do you mean with “pre-gestational DM”, it is a little confuse with the previous information of the first paragraph, please rephrase it to a single paragraph for further clarification. In addition I suggest add a reference to “Gestational DM affect 2-3% of pregnancy and pre-gestational DM affect 0.2-0.3% of pregnancy is the most common metabolic abnormalities of pregnancy which result in different adverse birth outcomes both for the mothers and neonates.”

#3 I also recommend to replace the term diabetic mother to a women with hyperglycaemic pregnancy.

#4 During all introduction section you referred to DM 1 and 2 and GDM, but your target is the hyperglycaemic pregnancies, so I think that is interesting bring some data separately in the first paragraph but in the others make sense to refer to this pregnancy condition as hyperglycemic disturbs (or any other similar expression).

# 5 In third paragraph the authors bring some interesting informations but I recommend rephrase it and use story telling technique to clarify and connect all the informations and paragraphs. In addition I suggest to delete the phrase “which is important for policy maker to set preventive strategies.” in the end of the introduction and add it to discussion section.

Methods session

# 6 Indicate if a review protocol exists, if and where it can be accessed (e.g., Web address), and, if available, provide registration information including registration number.

# 7 It is nuclear about the publication years of the articles selected, language, publication status

#8 I would like to recommend you to change the expression “poor quality” for the reason, I mean, which was the problem to not include it? Study design? Sample size? Not clear outcomes?

# 9 In phrase “the prospective follow up study reporting the prevalence of adverse birth outcomes or and a minimum of one contributing factors for adverse birth outcomes conducted in Ethiopia were included”. Did you establish a minimum and maximal follow up period? Let this information clear.

# 10 Please replace still births to stillbirths

# 11 Please replace soft wear to software in the phrase “The extracted data were transferred to STATA version 14 statistical soft wear for meta-analysis”

Results session

#12 In the first paragraph “Three hundred seventy seven articles were retrieved using a search strategy from PubMed, Cochrane Library, Google Scholar, SCOPUS, Web of Science and PsycINFO”, you didn’t mentioned the Addis Ababa and Haramaya University online repository.

# 13 I highly recommend to add the references into the tables to allow the reader find the articles easier.

#14 Table 4 was included Maternal age <35 years but in the text in socio-demographic factors session “Addis Ababa indicated that the odds of having adverse birth outcomes among diabetics women age less than 30 years old was 3.47” please clarify this point for us.

#15 Your results bring more than the purposed by the aim of your paper. “Therefore, this systematic review and meta-analysis is the first aimed to estimate the pooled prevalence of

adverse birth outcomes and associated factors among diabetic pregnant women in Ethiopia which is important for policy maker to set preventive strategies.” You actually brings the prevalence, pregnancy related risk factors and the adverse birth outcomes and during your data presentation in table 1, I expected that the pregnancy related risk factors should be present on it.

#16 Another concern is about the measure units in the table, please review it. I suggest to fix two decimal cases as standard to the table values.

Discussion Session

# 17 In order to improve the discussion I highly recommend the use of storytelling techniques to help you to delineate the findings and discuss it with the already included data. In the first phrase is interesting to add the mainly findings in an easy and quickly summarization to the reader follow the discussion topics of each findings and its possible explanation.

#18 It is of paramount importance to perform a major English revision

Conclusion Session

#19 The conclusion should address the aim.

AIM: “To estimate the pooled prevalence of adverse birth outcomes and associated factors among diabetic pregnant women in Ethiopia which is important for policy maker to set preventive strategies.”

Conclusion: “According to this systematic review and meta-analysis, the pooled prevalence of adverse birth outcome among diabetic women in Ethiopia was high. Unable to read and write, maternal age less than 30 years old, gestational age less than 37 completed weeks, previous adverse birth outcomes and no ANC follow up was significantly associated with adverse birth outcomes.”

I would like to rephrase the conclusion to further clarification. The phrase “Ethiopia was high” comparing to????. I suggest to replace by the %.

In addition, again I will insist that your data brings not only the prevalence of adverse birth outcomes, pregnancy related risk factors, but the adverse birth outcomes, and it should be addressed in the conclusion and in the aim.

Thank´s.

6. PLOS authors have the option to publish the peer review history of their article (what does this mean?). If published, this will include your full peer review and any attached files.

Reviewer #1: No

---

## [Author Response · Author response to Decision Letter 0]

14 Oct 2020

Point by point Author’s response to reviewer question and comment 

Editorial comment and suggestion: 

1. Please ensure that your manuscript meets PLOS ONE's style requirements, including those for file naming. The PLOS ONE style templates can be found at https://journals.plos.org/plosone/s/file? id=wjVg/PLOSOne_ formatting_ sample_main_body.pdf and https://journals.plos.org/plosone/s/file?i d=ba62/PLOSOne_ formatting_sample_title_authors_affiliations.pdf

Author’s response:-Sure! The manuscript has been formatted based on PLOS ONE's submission guideline include file naming. From repeated proof-reading of the manuscript, we found several grammatical errors, punctuation errors, wordings and spelling errors. Therefore, finding our colleague who has Master of Arts in English, we have tried our best to thoroughly copyedit the manuscript for English language usage. These changes are found throughout the revised version manuscript.

 Reviewer#1:

 Question and comments #1: I would like to congratulate the authors to address such a good topic to discussion. It is so important to bring to light the information about diabetes, which is an underestimated and poor screened disease during pregnancy. It is a condition which is relatively cheap to be detected and when in the majority of the cases have a good control with conservative approach but the implications of not address it is importance could affect negatively mother and children. All my comments are in order to improve the quality of the paper for the reader.

Tittle-The title is appropriate and brings curiosity for the reader to continue reading.

Author’s response: We are grateful for your appreciation of the educative being of our study and some editorial issue has been addressed in the title.

Question and comments #2: The abstract is appropriate; however, if you agree to change the points of the text, this must also be readjusted in the abstract.

Author’s response: We are agreed to change the pints of the text and has been readjusted accordingly.

Question and comments #3: The first paragraph of the introduction is a little bit complicated to understand; I suggest a major review of the English and rewrite it for better read.

Author’s response: After frequent proof-reading of the paragraph several grammatical errors, punctuation errors, wordings and spelling errors was there and the necessary revision was made. These changes are found in the manuscript.

Question and comments #4: In the second paragraph: what do you mean with “pre-gestational DM”, it is a little confuse with the previous information of the first paragraph, please rephrase it to a single paragraph for further clarification. 

Author’s response: Pre-gestational DM is replaced with the word Pre-existing DM. To avoid confusion the first and the second paragraph is rephrase to a single paragraph. The revisions are found in the manuscript. 

Question and comments #4: I suggest add a reference to “Gestational DM affect 2-3% of pregnancy and pre-gestational DM affect 0.2-0.3% of pregnancy is the most common metabolic abnormalities of pregnancy which result in different adverse birth outcomes both for the mothers and neonates.”

Author’s response: Definitely, the reference has been cited as reference # 3 and 4 found on introduction section of page 3, line 8 & 9. 

Question and comments #5: I also recommend replacing the term diabetic mother to a woman with hyperglycemic pregnancy.

Author’s response: Accepting and the term diabetic mother replaced with a woman with hyperglycemic pregnancy throughout the document.

Question and comments #6: During all introduction section you referred to DM 1 and 2 and GDM, but your target is the hyperglycemic pregnancies, so I think that is interesting bring some data separately in the first paragraph but in the others make sense to refer to this pregnancy condition as hyperglycemic disturbs (or any other similar expression).

Author’s response: We add sentence which referred about hyperglycemic pregnancies in introduction section on page 1, line 5 - 7.

Question and comments #6: In third paragraph the authors bring some interesting information but I recommend rephrase it and use story telling technique to clarify and connect all the information and paragraphs. In addition I suggest to delete the phrase “which is important for policy maker to set preventive strategies.” in the end of the introduction and add it to discussion section.

Author’s response: We tried to rephrase the paragraph accordingly and the phrase “which is important for policy maker to set preventive strategies” had been removed in the introduction section and added to discussion section. The revision is found in the manuscript.

 Methods session

Question and comments #7: Indicate if a review protocol exists, if and where it can be accessed (e.g., Web address), and, if available, provide registration information including registration number.

Author’s response: The review protocol was prepared before a review is started. It is registered at PROSPERO international data base with PROSPERO registration number (PROSPERO 2020: CRD42020167734). These revisions are found on the manuscript. 

Question and comments #8: It is not clear about the publication years of the articles selected, language, and publication status

Author’s response: Five articles published from year 2013-2019 and two unpublished articles in Addis Ababa and Haramaya University online repository reported in English language was selected for analysis. These revisions are found in the manuscript 

Question and comments #9: I would like to recommend you to change the expression “poor quality” for the reason, I mean, which was the problem to not include it? Study design? Sample size? Not clear outcomes?

Author’s response: It is to mean that article with JBI critical appraisal score less than 50% was excluded. The word “poor quality” is replaced with the word article with JBI critical appraisal score less than 50%

Question and comments #10: In phrase “the prospective follow up study reporting the prevalence of adverse birth outcomes or and a minimum of one contributing factors for adverse birth outcomes conducted in Ethiopia were included”. Did you establish a minimum and maximal follow up period? Let this information clear.

Author’s response: The minimum and maximum follow up period was not established. 

Question and comments #11: Please replace still births to stillbirths

Author’s response: The word still births has been replaced to stillbirths on page 8 (outcome measurement section).

Question and comments #12: Please replace soft wear to software in the phrase “The extracted data were transferred to STATA version 14 statistical soft wear for meta-analysis”

Author’s response- The word soft wear has been replaced to software on.

 Results session

Question and comments #13- In the first paragraph “Three hundred seventy seven articles were retrieved using a search strategy from PubMed, Cochrane Library, Google Scholar, SCOPUS, Web of Science and PsycINFO”, you didn’t mentioned the Addis Ababa and Haramaya University online repository.

Author’s response: Accepting the comment and the phrase Addis Ababa and Haramaya University online repository are added on page 9, line 3 (result section).

Question and comments #14: I highly recommend to add the references into the tables to allow the reader find the articles easier.

Author’s response: The references are cited in each included studies in table 1. 

Question and comments #15: Table 4 was included Maternal age <35 years but in the text in socio-demographic factors session “Addis Ababa indicated that the odds of having adverse birth outcomes among diabetics women age less than 30 years old was 3.47” please clarify this point for us.

Author’s response: it is type error and corrected as……women age less than 30 years old in table 4.

Question and comments #16: Your results bring more than the purposed by the aim of your paper. “Therefore, this systematic review and meta-analysis is the first aimed to estimate the pooled prevalence of adverse birth outcomes and associated factors among diabetic pregnant women in Ethiopia which is important for policy maker to set preventive strategies.” You actually brings the prevalence, pregnancy related risk factors and the adverse birth outcomes and during your data presentation in table 1, I expected that the pregnancy related risk factors should be present on it.

Author’s response: Pregnancy related risk factors are presented in table 1

Question and comments #17: Another concern is about the measure units in the table, please review it. I suggest to fix two decimal cases as standard to the table values.

Author’s response: The decimal case is changed to two decimal points in each table.

Discussion Session

Question and comments #18: In order to improve the discussion I highly recommend the use of storytelling techniques to help you to delineate the findings and discuss it with the already included data. In the first phrase is interesting to add the mainly findings in an easy and quickly summarization to the reader follow the discussion topics of each findings and its possible explanation.

Author’s response: The main finding of the study is written in the first phase of the paragraph and rearrangement was done; the second paragraph taken next to the first sentence and the paragraph next to the first sentence is deleted. The discussion is written based on an included data.

Question and comments #19: It is of paramount importance to perform a major English revision

Author’s response: Sure! From repeated proof-reading of the manuscript, we found several grammatical errors, interlinings, police titles, punctuation errors, wordings and spelling errors. Therefore, finding our colleague who has Master of Arts in English, we have tried our best to thoroughly copyedit the manuscript for English language usage. These changes are found throughout the revised version manuscript.

 Conclusion Session

Question and comments #20: The conclusion should address the aim.

AIM: “To estimate the pooled prevalence of adverse birth outcomes and associated factors among diabetic pregnant women in Ethiopia which is important for policy maker to set preventive strategies? ”Conclusion: “According to this systematic review and meta-analysis, the pooled prevalence of adverse birth outcome among diabetic women in Ethiopia was high. Unable to read and write, maternal age less than 30 years old, gestational age less than 37 completed weeks, previous adverse birth outcomes and no ANC follow up was significantly associated with adverse birth outcomes.”

I would like to rephrase the conclusion to further clarification. The phrase “Ethiopia was high” comparing to? I suggest to replace by the %.

In addition, again I will insist that your data brings not only the prevalence of adverse birth outcomes, pregnancy related risk factors, but the adverse birth outcomes, and it should be addressed in the conclusion and in the aim.

Author’s response: In conclusion section the word “high” replaced by 5.3%

The following statement (recommendation) was added based on the finding. “So, early screening of diabetes during pregnancy and follow up is important to alleviate the burden of maternal and neonatal adverse birth outcomes. In addition, concern should be given in improving antenatal follow up.” 

With regards!

Demeke Mesfin Belay (On behalf of all authors)

---

## [Decision Letter · Decision Letter 1]

21 Oct 2020

Adverse Birth Outcome and associated factors among Diabetic Pregnant Women in Ethiopia; Systematic review and Meta-analysis

PONE-D-20-26756R1

Dear Dr. Belay,

We’re pleased to inform you that your manuscript has been judged scientifically suitable for publication and will be formally accepted for publication once it meets all outstanding technical requirements.

Kind regards,

Frank T. Spradley

Academic Editor

PLOS ONE

Reviewers' comments:

Reviewer's Responses to Questions

**Comments to the Author**

1. If the authors have adequately addressed your comments raised in a previous round of review and you feel that this manuscript is now acceptable for publication, you may indicate that here to bypass the “Comments to the Author” section, enter your conflict of interest statement in the “Confidential to Editor” section, and submit your "Accept" recommendation.

Reviewer #1: All comments have been addressed

2. Is the manuscript technically sound, and do the data support the conclusions?

Reviewer #1: Yes

3. Has the statistical analysis been performed appropriately and rigorously? 

Reviewer #1: Yes

4. Have the authors made all data underlying the findings in their manuscript fully available?

Reviewer #1: Yes

5. Is the manuscript presented in an intelligible fashion and written in standard English?

Reviewer #1: Yes

6. Review Comments to the Author

Reviewer #1: The paper is appropriate and has been revised according to the reviewer's suggestions. The results are relevant and will contribute to the dissemination of knowledge in this important area of health.

7. PLOS authors have the option to publish the peer review history of their article (what does this mean?). If published, this will include your full peer review and any attached files.

Reviewer #1: No

---

## [Editor Report · Acceptance letter]

23 Oct 2020

PONE-D-20-26756R1 

Adverse Birth Outcome and associated factors among Diabetic Pregnant Women in Ethiopia; Systematic review and Meta-analysis 

Dear Dr. Belay:

I'm pleased to inform you that your manuscript has been deemed suitable for publication in PLOS ONE. Congratulations! Your manuscript is now with our production department. 

Kind regards, 

on behalf of

Dr. Frank T. Spradley 

Academic Editor

PLOS ONE